# Design and approval of the nutritional warnings' policy in Peru: Milestones, key stakeholders, and policy drivers for its approval

**Francisco Diez-Canseco**[1☺]*, **Victoria Cavero**[1☺], **Juan Alvarez Cano**[1☺], **Lorena Saavedra-Garcia**[1,2‡], **Lindsey Smith Taillie**[2,3‡], **Francesca R. Dillman Carpentier**[4‡], **J. Jaime Miranda**[1,5‡]

**1** CRONICAS Center of Excellence of Chronic Diseases, Universidad Peruana Cayetano Heredia, Lima, Peru, **2** Carolina Population Center, University of North Carolina at Chapel Hill, Chapel Hill, North Carolina, United States of America, **3** Department of Nutrition, Gillings School of Global Public Health, University of North Carolina at Chapel Hill, Chapel Hill, North Carolina, United States of America, **4** Hussman School of Journalism and Media, University of North Carolina at Chapel Hill, Chapel Hill, North Carolina, United States of America, **5** Department of Medicine, School of Medicine, Universidad Peruana Cayetano Heredia, Lima, Peru

☺ These authors contributed equally to this work.
‡ LSG, LST, FRDC and JJM also contributed equally to this work.
* francisco.diez.canseco.m@upch.pe

**Data Availability Statement:** Data cannot be publicly shared because it contains potentially identifying information. Since we interviewed key

## Abstract

Nutritional warnings are used as a public health strategy to address obesity. Peru approved in 2013 and implemented in 2019 a law requiring nutritional warnings on the marketing and packaging of processed foods high in sugar, sodium, saturated fat, and containing trans-fat. The complexity behind the design and approval of these policies over six years provides unique learnings, that inform the obesity prevention context, especially when proposed policies face strong opposition from powerful stakeholders. Our study aims to describe the milestones and key stakeholders' roles and stances during the nutritional warnings policy design in Peru, and to identify and analyze the main drivers of policy change that explain its approval. In 2021, interviews were conducted with 25 key informants closely involved in its design. Interviews were analyzed using the Kaleidoscope Model as a theoretical framework. Relevant policy documents and news were also analyzed. Milestones for this policy included the approval of the Law, Regulation, and Manual. Policy supporters were mainly from Congress, civil society advocates, and Health Ministers. Opponents came from Congress, ministries linked to the economic sector, the food industry, and media. Across the years, warnings evolved from a single text, to traffic lights, to the approved black octagons. Main challenges included the strong opposition of powerful stakeholders, the lack of agreement for defining the appropriate evidence supporting nutritional warning parameters and design, and the political instability of the country. Based on the Kaleidoscope Model, the policy successfully targeted a relevant problem (unhealthy eating decisions) and had powerful advocates who effectively used focusing events to reposition the warnings in the policy agenda across the years. Negotiations weakened the policy but led to its approval. Importantly,

stakeholders (e.g., former health ministers, congressmen and high-level civil servants) with specific and potentially unique knowledge about the nutritional warnings policy in Peru, the information they provided could ease their identification. The consent form we used in the study, stated: "any information that can identify you will never be shared publicly". Data requests could be sent to the corresponding author (fdiezcanseco@upch.pe) and the Institutional Research Ethics Committee (CIEI) of Universidad Peruana Cayetano Heredia (orvei.ciei@oficinas-upch.pe).

**Funding:** Funding was provided by Bloomberg Philanthropies (grant numbers 46129 and 2019-71181). The funders had no role in study design, data collection and analysis, decision to publish, or preparation of the manuscript.

**Competing interests:** The authors have declared that no competing interests exist.

government veto players were mostly in favor of the policy, which enabled its final approval despite the strong opposition.

## 1. Introduction

The prevalence of overweight and obesity in many Latin American countries is increasing more than 1% per year [1]. This rise has been associated with a greater availability of ultra-processed foods [2] high in sodium, sugars, and saturated fats, which increase the risk of obesity and related non-communicable diseases (NCDs) [3]. One strategy to address this problem is the use of nutritional warnings in processed and ultra-processed foods' front-of-package (FoP) labels [4], which have been recently implemented in Latin American countries, such as Chile, Peru, Uruguay, and Mexico [5] to inform the public about the high content of nutrients of concern, such as sugar, sodium, and fats [5].

Peru has a long history of undernutrition, but in the last decades, overweight and obesity increased in all age groups, reaching 38% of children in 2018 and 63% of adults in 2021 [6]. To address this problem, Peru launched the nutritional warnings policy, comprised of three main documents: the Law 30021, its Regulation, and the Nutritional Warnings Manual. In May 2013, the Peruvian Congress enacted Law 30021 called "Law for the Promotion of Healthy Eating for Children and Adolescents" [7] to reduce overweight and obesity and prevent NCDs in the future. The Law contains six strategies, including the use of nutritional warnings (later designed as black octagons) on the processed foods FoP labels and their publicity [7]. The Law delegated the design of a regulation to the Ministry of Health (MoH), based on recommendations of the World Health Organization (WHO) or the Pan American Health Organization (PAHO), to establish parameters for maximum levels of sugar, sodium, saturated fat, and trans-fat for processed foods and beverages. Processed foods were defined as industrially prepared, excluding minimally-processed foods [8] but including ultra-processed foods. If one or more of these parameters were surpassed, the corresponding nutritional warning would be used. The MoH had 60 days to prepare the regulation, which entailed a policy document to specify the Law's contents. The Law was finally approved four years later, in June 2017. The Regulation [8] specified two implementation dates of the warnings, with stricter nutritional parameters for the second one; and commissioned the MoH to prepare a "Nutritional Warnings Manual" to detail the warnings' design (e.g., shape, size, color, etc.). The Manual [9] was approved in June 2018, and the implementation of the nutritional warnings begun in June 2019, followed by the introduction of the stricter parameters in September 2021. This whole process lasted six years [10], a larger process compared to Chile [11] or Uruguay [12].

Importantly, in contrast to Chile [13] or Mexico [14, 15], whose policies processes have been studied and their main challenges, successes, and potential improvements described, the policy in Peru has received little attention. Documenting the Peruvian policy of nutritional warnings that are mandatory to comply, would be relevant for other countries who might implement similar policies. In this paper, we describe the milestones and key stakeholders' roles and stances during the design and approval of the nutritional warnings' policy in Peru (2013–2019) and analyze the policy drivers that might explain why it was finally designed and approved, despite the strong opposition that it faced over the years.

## 2. Methods

### 2.1 Study design

A qualitative study, consisting of semi-structured interviews with key stakeholders and a desktop review of news and policy documents related to the nutritional warnings' policy, was performed during 2021 to collect first-hand information about the process of designing and approving the nutritional warnings policy in Peru.

### 2.2 Participants

The informants were stakeholders, from different backgrounds and sectors, closely involved in the design, implementation and/or monitoring of the nutritional warnings policy in Peru. Our aim was to solicit the participation of up to 25 key informants, advocates, and opponents of the policy, from the following profiles: (1) policymakers (2) congressmen, (3) international organization representatives, (4) civil society advocates, (5) industry and media representatives, and (6) researchers. Some participants filled more than one role at different time-points (i.e., policymaker and civil society advocate). We proposed interviewing 9 policymakers representing the most diverse group of informants possible and with the greatest role in the policy design and implementation, plus 3 key informants from each of the other 5 profiles. A purposive sampling strategy [16] was used to select key informants who could provide detailed descriptions and valuable insights about the policy process from different perspectives. An initial group of informants was identified after reviewing the main policy documents and news, as well as from informal conversations with four of these informants, who helped us identify other key players. During each interview, new names emerged and were included in the list of potential informants. We finally interviewed 25 informants, as shown in Table 1.

### 2.3 Data collection tools

A semi-structured interview guideline was developed and organized into seven sections comprising questions about the design, implementation, and audit of the nutritional warnings policy in Peru. This comprehensive guideline was adapted to each interviewee, prioritizing those topics closer to their role and experience. New queries emerged during the data collection and were included in the following interviews if relevant.

**Table 1. Description of informants and number or interviews per profile.**

| Profile | Informants |
|---|---|
| Policymakers: Former or current Ministers, Vice-ministers and civil servants from different levels and Governmental institutions that were or are involved in the design, implementation and/or monitoring of the policy | 9 |
| Congressmen: Former congressmen involved in the design and approval of the Law and its Regulation, supporting or opposing the policy (based on votes, bills to modify the Law) | 3 |
| International organization representatives: Former and current representatives of international organizations informed about the design, implementation and/or monitoring of the policy | 3 |
| Civil society advocates: Former and current representatives of consumers associations or professional colleges involved in the design, implementation and/or monitoring of the policy | 4 |
| Industry and media representatives: Representatives from the food industry and media associations and from individual companies that publicly opposed or supported the policy | 3 |
| Researchers: Researchers and graduate students with backgrounds in social sciences and public health, whose research aimed to analyze discourses of the food industry during the design of this policy and continue conducting research in nutrition and policies in public health | 3 |
| **Total** | **25** |

For the desktop review, a rapid search using Google was conducted to retrieve news and policy documentation about the nutritional warnings policy to contextualize the interviewees' information and better understand the main policy contents and events. Search terms included "nutritional warnings," "octagons," "healthy eating law," and "Law 30021". News wase selected from Jan 1st, 2011, to June 30th, 2018, when the Manual was approved. Likewise, and using the Peruvian Government official website (www.gob.pe), the main policy documentations (i.e., the approved Law, Regulation and Manual, as well as previous and later drafts) were collected.

### 2.4 Procedures

Interviews were conducted in Spanish between January and September 2021 through video calls and were usually facilitated by two of the four researchers involved in the project since its conception. All interviews were audio recorded after obtaining informants' consent, and their average duration was 84 minutes (range: 53–118 minutes). When necessary, more than one session was arranged to cover all the interview topics.

### 2.5 Data analysis

Interview recordings were transcribed and a directed qualitative content analysis was performed [17] using ATLAS.ti 9. All interviews were double coded by two trained researchers, who lead or attended the interviews.

An initial codebook was developed based on the interview guideline's topics, with the aim of capturing essential milestones of the policy design and approval (i.e. approval of policy documents about the nutritional warnings), the role and stances of main stakeholders, and variables from the Kaleidoscope Model [18]. This model, designed to be used in LMICs such as Peru, provides a framework for analyzing policy-related changes. The model proposes a series of hypotheses that may explain the success of a given policy based on five stages of the political process: agenda setting, design, adoption, implementation, and evaluation and reform. We used the first three stages of the Kaleidoscope Model (agenda setting, design, and adoption), which entails a total of nine determinants and hypotheses, as shown in Table 2:

Before coding, the team met to standardize the procedure and understanding of the codes to be used. Along the coding process, emerging codes were discussed to decide if they would be added to the codebook. Disagreements were solved by discussions with another member of the team. The final codebook had 124 codes (S1 Text). This manuscript mainly presents the analysis of codes referring to the policy agenda setting, design, and adoption, from 2013 to 2019. The implementation and audit processes will be reported in future manuscripts.

Additionally, the most relevant policy documentation and news about the nutritional warnings' design and approval were used to contextualize and triangulate the informants' discourses and to prepare a timeline of main events. News and policy documentation were summarized by a researcher involved in the interview process using Excel matrices. These findings were then discussed with another researcher involved in the interview analysis and used to better understand the participants' discourses. These results mainly informed the timeline of events presented in Table 3.

### 2.6 Ethical considerations

The study was approved by Institutional Review Boards at Universidad Peruana Cayetano Heredia and University of North Carolina at Chapel Hill. All interviewees gave informed consent. Some signed a virtual form while others gave oral consent during a recorded phone call.

**Table 2. Components of the Kaleidoscope Model, including policy stages and its determinants of policy change.**

| Policy stage | Determinant of policy change | Hypothesis to explain how determinants explain the policy change |
|---|---|---|
| Agenda setting: How the policy reached the policy agenda | Recognized, relevant problem | Hypothesis 1: The policy reached the agenda because advocates recognized and proposed a relevant problem, based on credible evidence or popular perception |
| | Focusing event | Hypothesis 2: The policy reached the policy agenda because a well-defined event focused public attention on a problem or created a window of opportunity for the policy agenda setting |
| | Powerful advocates | Hypothesis 3: The policy reached the policy agenda because strong individuals, organizations, or companies supported a new or changed policy to key decision makers. |
| Design: How the policy's specific design was defined | Knowledge and research | Hypothesis 4: The policy was designed using evidence-based knowledge that shaped feasible design options. |
| | Norms, biases, ideology, and beliefs | Hypothesis 5: The policy was designed based on beliefs and biases that shaped the range of design features that were acceptable |
| | Cost-benefit calculations | Hypothesis 6: The policy was designed based on expected costs and expected benefits (i.e. which interests will gain and which will lose as a result of different designs; which design is more affordable) that determined the preferred design |
| Adoption: How the policy was issued or signed | Powerful opponents vs proponents | Hypothesis 7: The policy was adopted because supporters were relatively more powerful than opponents. |
| | | The policy was not adopted because opponents were relatively more powerful than supporters |
| | Government veto players | Hypothesis 8: The policy was adopted because government agents with ultimate decision-making power were supportive or neutral of the policy. |
| | | The policy was vetoed because government agents with ultimate decision-making power opposed the policy |
| | Propitious timing | Hypothesis 9: The policy was adopted because supporters waited for opportune moments (political, economic, social) to push the policy approval |

Source: Haggblade & Babu (2017) [19]

## 3. Results

We present the milestones during the nutritional warnings' design by describing the processes of designing and approving the Law 30021, its Regulation, and the Manual (Table 3), as well as the role of key stakeholders along the years (Figs 1–3). We also outline how the nutritional warnings' parameters and design were defined and summarize the main challenges faced during these processes. Based on this narrative, we present a table using the Kaleidoscope Model to analyze how and why this policy was ultimately approved, despite the long periods and strong opposition.

### 3.1 In 2013, nutritional warnings in FOP labels and publicity are included in the Law 30021

**3.1.1 Summary of how the Law was designed and approved.** According to our informants, the nutritional warnings in Peru aimed to reduce obesity and promote healthy eating habits in children and adolescents by 1) informing the population about the high content of

**Table 3. Timeline of main events related to the nutritional warnings policy's design and approval in Peru, 2011–2019.**

| Year | Policy Events | Political Events | Scientific Events | Other Events |
|---|---|---|---|---|
| 2011 | | • Jul 27: The new elected Congress begins their activities [20] | • 2011: Report: Recommendations of the PAHO Expert Consultation on the Promotion and Advertising of Foods and Non-alcoholic Beverages to Children in the Region of the Americas [21] | • PAHO: May 12–13: Expert Consultation on the Promotion and Advertising of Foods and Non-alcoholic Beverages to Children in the Region of the Americas, convened by PAHO/WHO [21] |
| | Jul 28: Ollanta Humala assumes the presidency [22] (Party: Peruvian Nationalist Party, majority in Congress—47 congressmen out of 130) [20] | | | |
| | | | | • Peru: Nov 17–19: 5th National Health Conference [23]<br>• Peru: Dec 30: Civil society organization's (Foro Salud) Policy Statement at the 5th National Health Conference asking for regulations to prevent NCDs [24] |
| 2012 | • Jan 30: First bill (Law draft) 774/2011-CR [25]<br>• Jan 30: Second bill 775/2011-CR [26]<br>• Apr 23: Third bill 1038/2011-CR [27] | • Jul 23: Dr. Midori de Habich becomes new Minister of Health [28] | • Jan 23: Study of unhealthy food advertising on Peruvian television—CONCORTV [29] | • Chile: Jun 06: Enactment of Law 20606 in Chile—"About the nutritional composition of foods and their advertising" [30] |
| 2013 | **17/05: Enactment of the Healthy Eating Law (Law 30021) [7]** | | | |
| 2014 | • Mar 24: Prepublication of the first Regulation's draft [31] | • Nov 05: Dr. Anibal Velásquez becomes new Minister of Health [32] | • Journal Article: May: "Percentile values for sugar, fat and sodium content in industrialized foods according to labeling issued in Lima" [33] | • Ecuador: August: Approval of the "Substitute Sanitary Regulation for Processed Foods for Human Consumption 5103", which mandates the use of traffic lights [34]<br>• PAHO: Sep 29 –Oct 03: 53rd PAHO Directing Council—approval of the action plan for the prevention of obesity in childhood and adolescence (Requests PAHO to provide technical parameters based on scientific evidence) [35] |
| 2015 | • Mar 20: Bill 4343/2014-CR bill to modify Law 30021 [36]<br>• Apr 17: Approval of a Regulation using the technical parameters of PAHO's Expert Recommendations (never implemented) [37]<br>• Sep 10: Bill 4808/2015-CR bill to modify Law 30021 [38] | | • Report: WHO Regional Office for Europe—Nutrient Profile Model [39] | • Chile: Jun 26: Regulation of Law 20606—Establishment of technical parameters and black octagons as warnings [40]<br>• Peru: May 18: "Healthy Eating Walk" led by the College of Nutritionists of Peru [41] |
| 2016 | • Feb 25: Ministry of Justice's Dictum regarding the Law 30021 scope<br>• Jul 25: Prepublication of the second Regulation's draft [42]<br>• Jul 27: Approval of Trans-Fat's Regulation [43] | • Jul 27: The new elected Congress begins their activities [44] | • Report: PAHO Nutrient Profile Model [45] | |
| | Jul 28: Pedro Pablo Kuczynski assumes the presidency (Party: Peruvians for Change; opponent party *Fuerza Popular* has majority in Congress -73 congressmen out of 130) [46] | | | |
| | | • Jul 28: Patricia García becomes new Minister of Health [47] | | |

*(Continued)*

**Table 3.** (Continued)

| Year | Policy Events | Political Events | Scientific Events | Other Events |
|---|---|---|---|---|
| 2017 | • Jan 11: Bill 865/2016-CR, bill to modify Law 30021 [48]• Jun 12: Bill 1519/2016-CR, bill to modify Law 30021 [49]<br>• **Jun 17: Approval of the Regulation of the Law 30021 [8]**<br>• Jun 23: Bill 1589/2016-CR, bill to modify Law 30021 [50]• Jul 24: Bill 1700/2016-CR, bill to modify Law 30021 [51]• Aug 17: Prepublication of the first draft of the Nutritional Warnings' Manual [52]• Oct 04: Bill 1959/2017-CR bill to modify Law 30021 [53]• Oct 19: Bill 2036/2017-CR bill to modify Law 30021 [54]• Nov 17: The Congress' Working group approves the Dictum to modify the Law 30021 [55] | • Sep 17: Fernando D´Alessio becomes new Minister of Health [56]<br>• Dec 21: First impeachment motion to President Kuczynski [57]<br>• Dec 24: President Kuczynski's pardon to incarcerated and former dictator Alberto Fujimori [58] | | • Panama: Jun 02: "Pura Vida" scandal: Panama stops commercialization of the dairy product "Pura Vida" [59]<br>• Peru: October: Presentation of "NutriAPP" app [60]<br>• Peru: Jun 19: Protest by civil society led by the College of Nutritionists of Peru against the Regulation of Law 30021 [61]<br>• Peru: Sep 20: Demand of the College of Nutritionists of Peru to MoH against the approved technical parameters [62] |
| 2018 | • Mar 09: Approval of Dictum that modifies the Law 30021 [63]• Mar 28: Congressmen send the approved Dictum to the President [64]<br>• **Jun 16: Approval of the Nutritional Warnings' Manual [9]** | • Jan 09: Abel Salinas becomes new Minister of Health [65]<br>• Mar 21: Kuczynski resigns as president [66] | | |
| | Mar 23: President Kuczynski resigns and Martin Vizcarra, his vice-president, is elected president [67] | | | |
| | | • Apr 02: Silvia Pessah becomes new Minister of Health [68]<br>• June: Keiko Fujimori (*Fuerza Popular*' leader and daughter of Alberto Fujimori) meets with President Vizcarra to ask him not to approve the octagons as design for the nutritional warnings' design [69] | | |
| 2019 | • May 17: Publication of Questions & Answers to the industry about the Nutritional Warnings' Manual [70]<br>• Jun 17: The first implementation phase begins [71] | | | |

nutrients of concern in processed food; 2) encouraging the consumption of healthier options (with fewer or no warnings); and 3) encouraging the food industry to improve the nutritional content of their products. These warnings reached the Peruvian policy agenda as one of the six components of the "Healthy Eating Law" (Law 30021) [7].

This Law was the result of three bills (Law drafts) designed by congressmen from different parties between January and April 2012 [25–27]. These bills were merged into one single text and discussed in the Congress for several months, when the text was constantly edited mainly based on critics from media and food industry representatives but also on some recommendations from public institutions, civil society advocates, and PAHO [72]. Despite the strong opposition and influence of food industry and media representatives, the majority of the 120 congressmen were in favor of the Law, which enabled its enactment and further endorsement by the country President, President Humala (2011–2016), in May 2013. The constant editions removed important elements that many interviewees regretted, but they also commented that it was the best that could be achieved after all the opposition received.

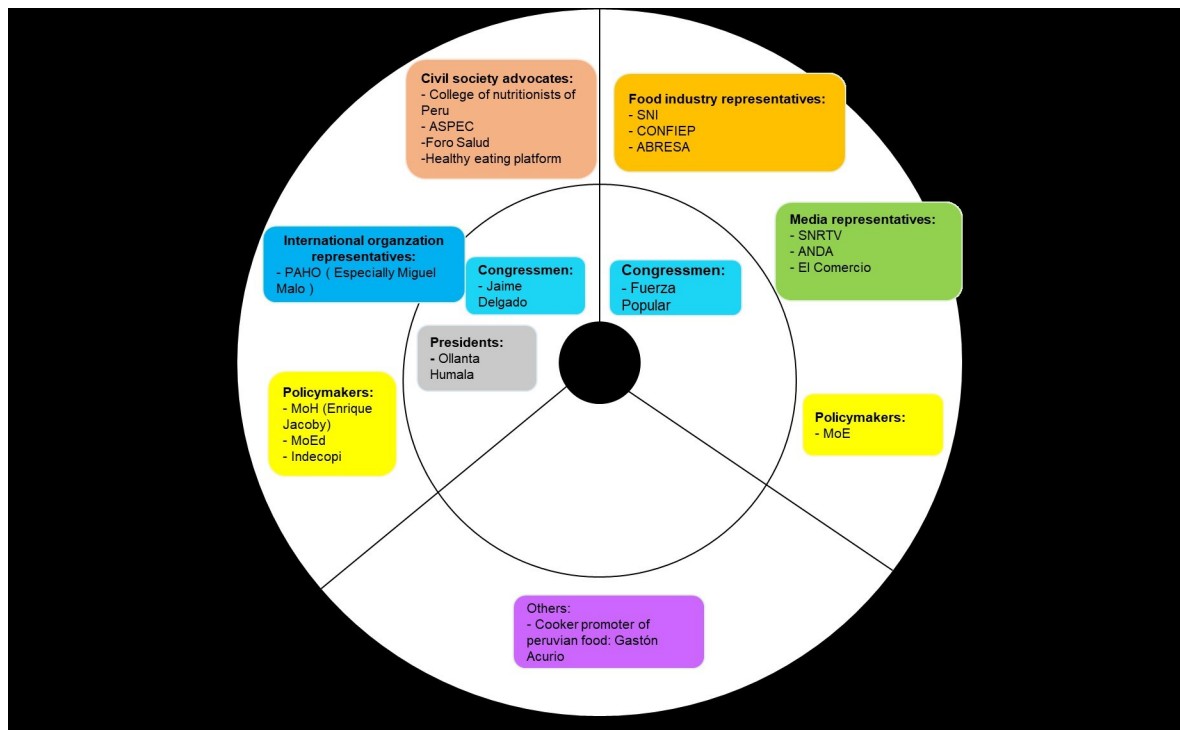

**Fig 1. Stances of key stakeholders during the Law 30021's design and approval (2012–2013).** Stakeholders in the inner circle are those with higher power in the decision-making process. <u>Abbreviations</u>: Ministry of Health (MoH), Ministry of Economy and Finances (MEF), Ministry of Education (MoE), Ministry of Justice (MoJ), Ministry of Social Inclusion (MoS), Ministry of Foreign Trade and Tourism (MoF), Ministry of Agriculture (MoA), Ministry of Production (MoP), National Radio and Television Society (SNRTV), National Advertisers Association (ANDA), National Confederation of Private Entrepreneurial Institutions (CONFIEP), National Society of Industries (SNI), Non-alcoholic Beverage and Soft Drink Industry Association (ABRESA), Peruvian Association of Consumers and Users (ASPEC).

"The congressman who was in charge told me 'Ms. that's what was possible, this is the Law that we were able to do'. (. . .) So, yes, that left me thinking a lot. I then began to say: 'The Law is not perfect, but it is the only thing, the best we have been able to achieve and with this imperfect Law we can now move forward´´´. (Civil society advocate, SOC-01)

**3.1.2 How were the nutritional warnings' parameters defined during the design period of designing the Law?.** Nutritional warnings required parameters to define the maximum nutrients' content that a processed product could have to avoid the warning in its FOP label and publicity. The third bill (before the Law) [27] proposed using a set of parameters produced during a PAHO expert consultation in 2011 [21], which established maximum contents of sugar, fat, sodium, and trans-fat for processed foods (Table 4). However, opponents argued that these parameters were based on experts' opinions and not on an official PAHO document nor scientific evidence. Thereby, the Law did not include any parameter but instead commissioned the MoH to define them in a further Regulation [7]. This Regulation should be published in no more than 60 days and parameters should be "based on recommendations from WHO/PAHO".

**3.1.3 Proposed nutritional warnings in the Law.** The Law did not mention any specific design for the nutritional warnings but only the following texts: "High in [sodium/sugar/fat]: Avoid its excessive consumption" and "Contains trans-fat: Avoid its consumption," which

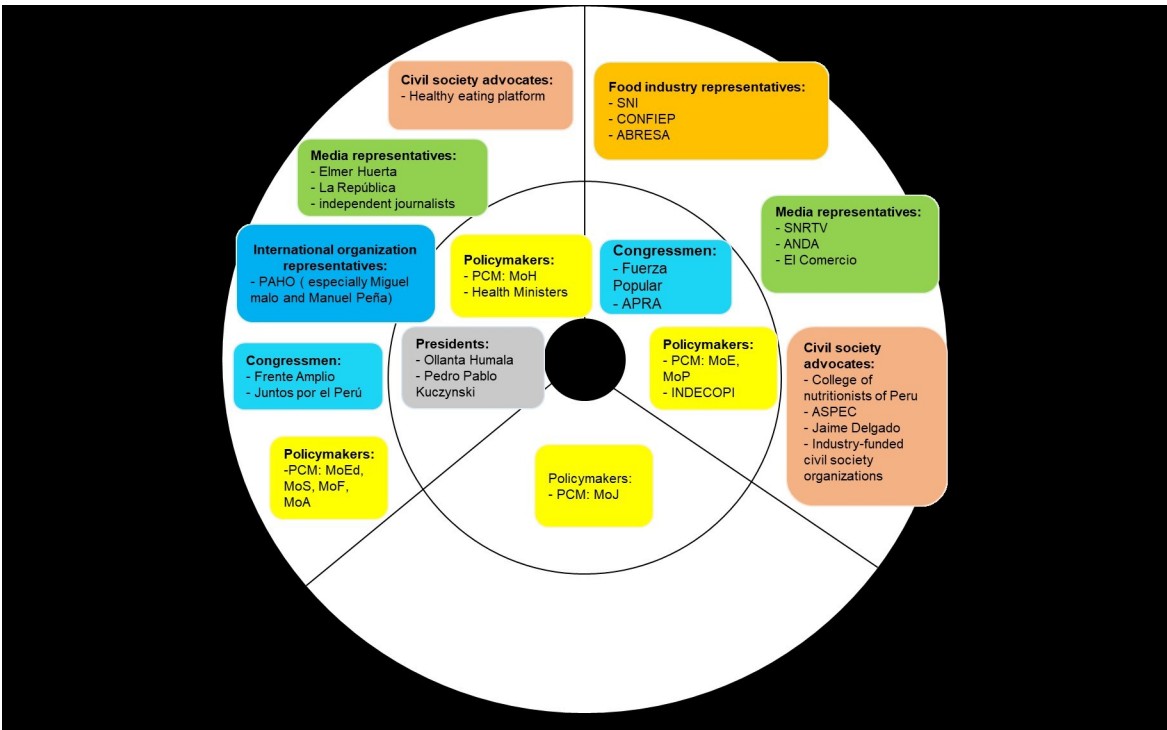

**Fig 2. Stances of key stakeholders during the Regulation's design and approval (2014–2017).** Stakeholders in the inner circle are those with higher power in the decision-making process. <u>Abbreviations</u>: Ministry of Health (MoH), Ministry of Economy and Finances (MEF), Ministry of Education (MoE), Ministry of Justice (MoJ), Ministry of Social Inclusion (MoS), Ministry of Foreign Trade and Tourism (MoF), Ministry of Agriculture (MoA), Ministry of Production (MoP), National Radio and Television Society (SNRTV), National Advertisers Association (ANDA), National Confederation of Private Entrepreneurial Institutions (CONFIEP), National Society of Industries (SNI), Non-alcoholic Beverage and Soft Drink Industry Association (ABRESA), Peruvian Association of Consumers and Users (ASPEC).

should be clearly seen in the products' FoP labels and publicity [7], similar to the warnings used in tobacco. See Fig 4 to appreciate the different designs proposed across time and Table 5 to understand how the warning designs changed from traffic light label to a warning label.

**3.1.4 Main challenges faced when trying to approve the Law 30021.** The main challenges in this period were 1) the strong opposition of the food industry and media representatives to some aspects of the Law and 2) the little scientific evidence to define the nutritional warnings' parameters. Arguments used by those in favor and against the warnings can be found in Table 5. Importantly, some informants said that during this period opponents were mainly focused on the regulations to publicity (another component of the Law) rather than on the nutritional warnings, arguing for instance that the Law was against media freedoms. However, this strong opposition risked the approval of the Law.

> "People from the Society of Radio and TV said: 'You know what, this project can't be approved with that [regulations to publicity]. We are not interested if you put labels, whatever. . . [but] don't mess with publicity'. The same day, someone from the Fujimori's party, comes and says, 'If you don't take that out, that article, don't even count with our votes'. And then others from another party, who were supposedly my allies, 'If you don't take that out, there is no Law'. And I'm like 'Wow, how much power!'" (Congressman 2011–2016, POL-02).

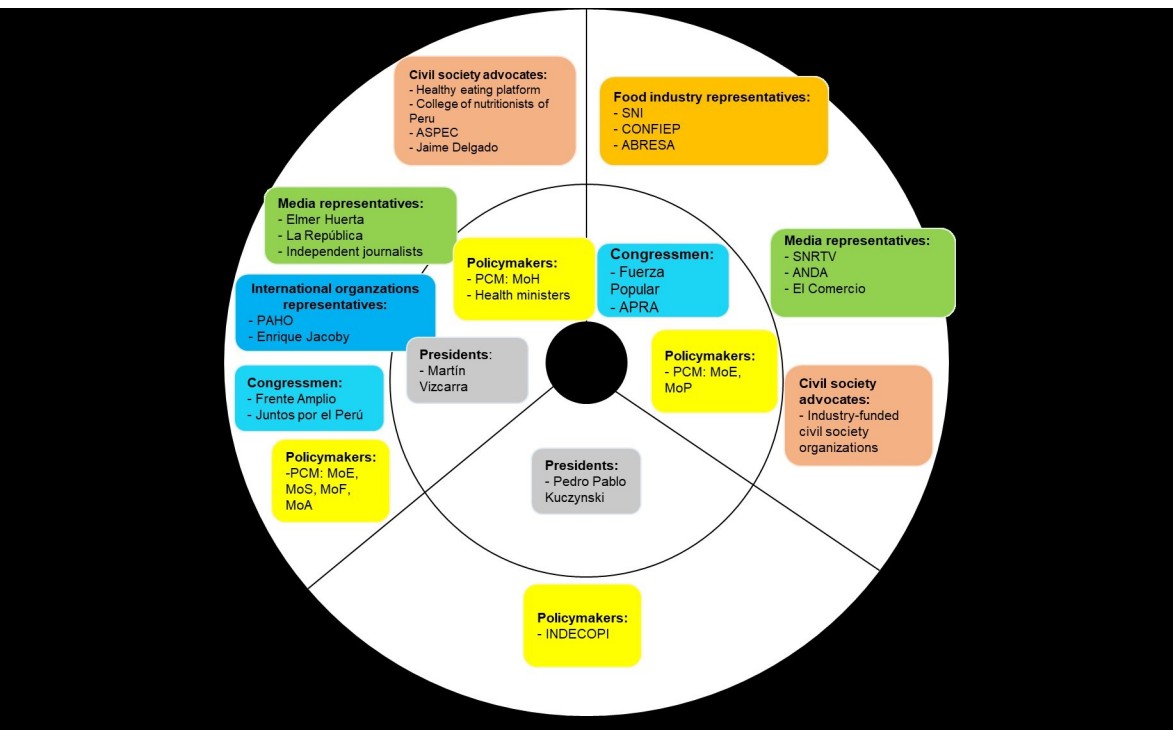

**Fig 3. Stances of key stakeholders during the Nutritional Warnings Manual design and approval (2017–2018).** Stakeholders in the inner circle are those with higher power in the decision-making process. Abbreviations: Ministry of Health (MoH), Ministry of Economy and Finances (MEF), Ministry of Education (MoE), Ministry of Justice (MoJ), Ministry of Social Inclusion (MoS), Ministry of Foreign Trade and Tourism (MoF), Ministry of Agriculture (MoA), Ministry of Production (MoP), National Radio and Television Society (SNRTV), National Advertisers Association (ANDA), National Confederation of Private Entrepreneurial Institutions (CONFIEP), National Society of Industries (SNI), Non-alcoholic Beverage and Soft Drink Industry Association (ABRESA), Peruvian Association of Consumers and Users (ASPEC).

### 3.2 In 2017, the parameters for the nutritional warnings are finally defined in the Regulation of the Law 30021

**3.2.1 Summary of how the Regulation was designed and approved.** The Law's Regulation was approved four years later than expected, in June 2017 [8]. During these years, the MoH pre-published two Regulation drafts [42, 73] with different parameters for sugar, salt, and saturated fat in each, as shown in Table 4. Drafts had to be pre-published for 90 days to receive opinions before its official approval. For each draft, the MoH assigned two working groups to design the Regulation: the sectoral working group, composed only by the MoH, in charge of defining the nutritional warnings' parameters; and the multisectoral working group, composed of representatives from different ministries and Indecopi, to design the Regulation for all the Law.

Simultaneously, congressmen against the Law proposed four bills to modify it in 2015 [36, 38] and 2017 [48, 49]. Informants from the MoH commented on the pressure and challenging atmosphere of having to design the Regulation while having Congress attempting to change the Law. Also in these years, civil society advocates led protest marches and were in news programs to demand the implementation of the Law [41]. Our informants believed that these actions increased public awareness and pressed the MoH to accelerate their work.

In June 2017, a large Peruvian company of dairy products was sanctioned for advertising a product named "Pura Vida" as milk when it was a beverage made of milk and vegetable

**Table 4. Timeline of parameters for maximum levels of sugar, saturated fat, sodium, and trans-fat in processed foods and beverages over the years, including those proposed by WHO/PAHO and those used in the nutritional warnings' policy documents.**

| Year | Documents | Phases | Food category | Total sugar | Saturated fat | Sodium | Energy | Trans fat |
|---|---|---|---|---|---|---|---|---|
| 2008 | Trans Fat Free Americas—Rio de Janeiro Declaration* | Single phase | Oils and margarines | | | | | < 2% total fat |
| | | | Processed foods | | | | | < 5% total fat |
| 2011 | Recommendation of the PAHO expert consultation on the promotion and advertising of food and nonalcoholic beverages to children in the Region of the Americas* | Single phase | Solid food | ≥ 5g/100g | ≥ 1.5g/100g | ≥ 300mg/100g | | > 0 gr |
| | | | Beverages | ≥ 2.5g/100ml | ≥ 0.75g/100ml | ≥ 300mg/100ml | | > 0 gr |
| 2014 | First draft of the Regulations of Law N˚30021 (Based on CENAN study) (RM 321-2014/MINSA)** | Single phase | Non-alcoholic beverages | ≥ 8.1g/100ml | ≥ 5.3g/100ml | ≥ 540mg/100ml | | Fats, vegetable oils and margarines: ≥ 2g/100g or 100ml of fatty matter |
| | | | General solid foods | ≥ 12.5g/100g | ≥ 5.3g/100g | ≥ 540mg/100g | | |
| | | | Grains and derivatives | ≥ 12.5g/100g | ≥ 3.4g/100g | ≥ 540mg/100g | | Other industrially processed food and non-alcoholic beverages: ≥ 5g/100g or 100ml of fatty matter |
| | | | Cakes, biscuits and cookies | ≥ 19.6g/100g | ≥ 9.8g/100g | ≥ 540mg/100g | | |
| | | | Snacks | ≥ 12.5g/100g | ≥ 7g/100g | ≥ 540mg/100g | | |
| 2015 | Regulations technical parameters–MINSA (DS 007-2015-SA)** | Single phase | Solid food | ≤ 5g/100g | ≤ 1.5 g/100g | ≤ 300 mg/100g | | |
| | | | Beverages | ≤ 2.5g/100ml | ≤ 0.75g/100ml | ≤ 300 mg/100ml | | |
| | Chilean technical parameters* | 1st phase | Solid food | > 22.5g/100g | > 6g/100g | > 800mg/100g | > 350 Kcal/100g | |
| | | | Beverages | > 6g/100ml | > 3g/100ml | > 100mg/100g | > 100 Kcal/100ml | |
| | | 2nd phase | Solid food | > 15g/100g | > 5g/100g | > 500mg/100g | > 300 Kcal/100g | |
| | | | Beverages | > 5g/100ml | > 3g/100ml | > 100mg/100g | > 80 Kcal/100ml | |
| | | 3rd phase | Solid food | > 10g/100g | > 4g/100g | > 400mg/100g | > 275 Kcal/100g | |
| | | | Beverages | > 5g/100ml | > 3g/100ml | > 100mg/100g | > 70 Kcal/100ml | |

(*Continued*)

**Table 4.** (Continued)

| Year | Documents | Phases | Food category | Total sugar | Saturated fat | Sodium | Energy | Trans fat |
|------|-----------|--------|---------------|-------------|---------------|--------|--------|-----------|
| 2016 | Pan American Health Organization nutrient profile model* | Single phase | Solid food | ≥ 10% of total energy | ≥ 10% of total energy | ≥ 1 mg/1 Kcal | | ≥ 1% of total energy |
| | | | Beverages | ≥ 10% of total energy | ≥ 10% of total energy | ≥ 1 mg/1 Kcal | | ≥ 1% of total energy |
| | Second draft of the Regulations to Law N° 30021 (RM 524-2016/MIINSA)** | Single phase | Solid food | ≥ 10% of total Kcal | ≥ 10% of total Kcal | ≥ 1 mg/ Kcal | | |
| | | | Beverages | ≥ 10% of total Kcal | ≥ 10% of total Kcal | ≥ 1 mg/ Kcal | | |
| | Trans fat regulation (DS 033-2016-SA)** | 1st phase | Fats, vegetable oils and margarines | | | | | ≥ 2g/100g o 100ml of fatty matter |
| | | | Other industrially processed food and non-alcoholic beverages | | | | | ≥ 5g/100g o 100ml of fatty matter |
| | | 2nd phase | All processed foods | | | | | Elimination of the use and content of trans fats from partial hydrogenation |
| 2017 | Regulation of Law N° 30021 (DS 017-2017-SA)** | 1st phase | Solid food | ≥ 22.5g/ 100g | ≥ 6g/100g | ≥ 800 mg/100g | | |
| | | | Beverages | ≥ 6g/ 100ml | ≥ 3g/100ml | ≥ 100mg/ 100ml | | |
| | | 2nd phase | Solid food | ≥ 10g/ 100g | ≥ 4g/100g | ≥ 400mg/ 100g | | |
| | | | Beverages | ≥ 5g/ 100ml | ≥ 3g/100ml | ≥ 100mg/ 100g | | |

* Parameters published by WHO/PAHO

** Parameters used in the Peruvian policy documents

products [74]. This case received great media coverage and the proponents of the Law advocated for the Regulation's approval arguing that consumers should be properly informed about what they are eating, which was the purpose of the nutritional warnings. Civil society advocates led protests and made media appearances, pressing the newly elected government to approve the Regulation [61]. On June 15th, 2017, two weeks after the "Pura Vida" scandal, the Regulation was approved.

"Without "Pura Vida" we would not have a Regulation. I am completely sure of that." (Decisionmaker at the MoH, DEC-07)

The approved Regulation used the parameters of Chile and commissioned the MoH to design and approve a Manual with the technical characteristics of the nutritional warnings in no more than 120 days.

**3.2.2 How the nutritional warning parameters were defined in the period of designing the Regulation.** Once the Law was approved, the MoH oversaw the design of its Regulation.

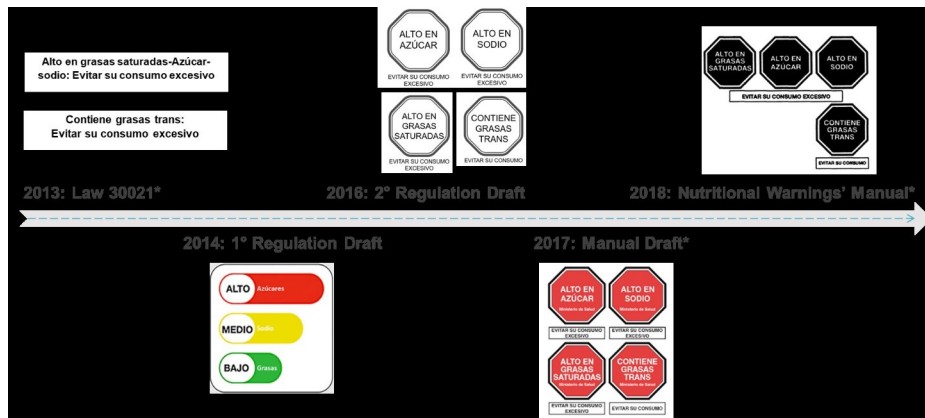

**Fig 4. Nutritional warnings' designs proposed during the main policy documents design.** *Only these designs were published in official documents. The Law specified the text for the nutritional warnings, similar to the ones used in tobacco. When designing the first Regulation Draft, its authors suggested using the traffic lights design, similar to the ones being used in Ecuador at that time. Since the second Regulation Draft, the authors suggested using the octagons design, but with some differences, such as color and shape.

During these years, the food industry proposed the daily reference values defined by the CODEX Alimentarius [72], which is a set of guidelines and standards approved by the WHO. The MoH, however, did not recommend their use due to their basis on the nutritional requirements of adults and not children or adolescents, and proposed different parameters along the years (Table 4).

In 2014, the MoH pre-published a Regulation's draft with parameters for sugar, sodium, and fat, based on an existing database of only 400 processed foods and beverages collected in July 2013 [73]. This draft was strongly criticized for using parameters too flexibly and for not being based on the WHO/PAHO recommendations, as indicated in the Law.

> "Someone who criticized this Regulation a lot was Manuel Peña [former PAHO representative]. He created a strong pressure when he said in the media that this Regulation was not the one that was [proposed] in the Law, that the values [of this draft] were not the ones specified in the Law" (Decision maker at the MoH, DEC-01)

In 2015, and with a new Minister of health, the MoH held several meetings with PAHO in Peru and officially asked for their parameters. Our informants commented that PAHO replied by sending the parameters from the PAHO's expert consultation of 2011, which was used as evidence by the sectoral commission to publish a Regulation with these parameters [74], in an attempt to cut off discussions at the multisectoral working group that might have led to the adoption of less stringent parameters.

> "We could not issue a new version of the Regulation if it was not done in a multisectoral working group, and that would have taken a long time. So, we decided that the MoH defines the parameters, which were the most critical point (. . .) (and) The other things, (for example) the label's design or the timeframe, all that, was part of the Regulation. But the value would not be in dispute. So, (with this publication) what was avoided was the negotiation of parameters. They would be already established." (Decision maker at the MoH, DEC-01)

Some months later, Chile issued its own parameters for their nutritional warnings [40], which were more flexible than those of the expert consultation but more stringent than those

**Table 5. Main arguments used by supporters and opponents of the nutritional warnings policy and how the arguments were resolved.**

| Topic | Arguments against | Arguments in favor | How it was resolved |
|---|---|---|---|
| **Include regulations to publicity** | • **Design of the Law:**<br>• The law goes against international agreements and because of that they were going to impose sanctions on Peru.• Soccer games and the World Cup will not be broadcasted because they have processed food sponsors.• Regulations come with political and ideological overtones against companies.• Advertising companies have signed a self-regulation agreement.• The law is an encroachment on the media or is against media freedoms. | • **Design of the Law:**<br>• Overweight and obesity in children and adolescents must be addressed because it is a serious problem.• The law seeks to protect children and adolescents because they are a vulnerable population.• Foods that are harmful to health should not be advertised.• Advertising of ultra-processed foods influences children and adolescents to buy them.• The industry invests a lot of money in advertising ultra-processed foods, so the industry is against this law. | • **Design of the Law:**<br>• The ban on advertising unhealthy processed foods in family protection time (6 am to 10 pm) was removed.• Several articles were eliminated or modified (e.g. the prohibition to use animated characters in advertising was eliminated). |
| **Choice of nutritional warnings** | • **Design of the Law:**<br>• It will have a negative impact on the economy and job losses in the food industry. T• here is no scientific evidence that ultra-processed foods will make people sick or kill them.• The problem is not ultra-processed foods but eating habits.• There are no bad foods.• Nutritional warnings will be placed on typical Peruvian dishes. | • **Design of the Law:**<br>• Overweight and obesity in children and adolescents must be addressed because it is a serious problem.T• he consumption of ultra-processed foods and chronic diseases have increased due to a poor diet in Peru.• We are not prohibiting the sale of any food but ensuring the availability of information to consumers.• Nutritional warnings will help the food industry to improve the quality of their products. | • **Design of the Law:**<br>• Nutritional warnings were maintained |
| | • **Design of the 2nd draft of the regulation:**<br>• The Law targets children and adolescents exclusively, so only ultra-processed foods targeted for children and adolescents must have nutritional warnings• The Regulation cannot include the iconography of the nutritional warnings since the Regulation cannot restrict more than the Law and include contents that it did not specify.• Processed foods only represent 9.1% of Peruvians' food consumption. | • **Design of the 2nd draft of the regulation:**<br>• The Regulation should expand the contents of the Law in order to achieve its objectives and purpose, which is to protect, warn and maintain a healthy diet• Scientific evidence shows that iconography helps to achieve the Law's goal.• The Regulation should apply to the entire population, but with emphasis on children and adolescents. | • **Design of the 2nd draft of the regulation:**<br>• The Ministry of Justice issued an official letter stating that: 1) The Regulation should have an emphasis on children and adolescents; and 2) The regulation cannot consider the size, color and iconography of the nutritional warnings.• The Ministry of Health prepares a regulation with emphasis on children and adolescents and does not regulate the iconography of nutritional warnings, but it does leave a project where it indicates that for them, the nutritional warnings should be octagons. |
| | • **Design of the Nutritional Warnings Manual:**<br>• Octagons do not provide all the necessary information, so a combination of the GDA and the nutritional traffic light is better.• Not providing all the information to the consumer and only displaying seals such as octagons was going to stultify the consumer.• The congress proposal was to promote and disseminate the nutritional tables and the combination of the GDA-Traffic lights.• The red octagon is an obstacle to trade.• The food industry opposed placing "Ministry of Health" on the octagons. | • **Design of the Nutritional Warnings Manual:**<br>• The octagon is easier for the consumer to understand and the congress proposal (GDA-Traffic lights) is very confusing.• There is scientific evidence that the octagon is better than the traffic light.• There were lobby for the traffic lights. | **Design of the Nutritional Warnings Manual:**<br>• President Vizcarra observes the Law approved by the congress to implement the GDA-Traffics Light, and this, together with social and media pressure, manages to stop the congressional attempt to modify the law.• The congress does not continue with the discussion of the modifications, even though they were not in favor of the octagons, because it was already necessary to start implementing a Law regulating processed foods.• The Ministry of Health approves the nutrition warnings manual with the black octagon nutrition warning and removes the text "Ministry of Health". |

(*Continued*)

**Table 5.** (Continued)

| Topic | Arguments against | Arguments in favor | How it was resolved |
|---|---|---|---|
| **Choice of technical parameters of sugar, sodium, and saturated fat** | • **Design of the Law:**<br>• The technical parameters have no scientific evidence and are not in an official PAHO document. | • **Design of the Law:**<br>• If the technical parameters are removed, it will take years to define them. | • **Design of the Law:**<br>• The technical parameters of the Law are withdrawn so that they can be discussed in the Regulation.• The Law states that the Regulation should be based on WHO/PAHO recommendations. |
| | • **Design of the 1st draft of the regulation:**<br>• The technical parameters are not based on WHO/PAHO recommendations.• The technical parameters are not based on what is ideal for ultra-processed foods, but on what is already on the market.• The technical parameters are very restrictive and lack scientific evidence.• Ultra-processed foods only represent 9.1% of Peruvians' food consumption. | • **Design of the 1st draft of the regulation:**<br>• There is no scientific evidence or any official PAHO document that indicates the technical parameters, so technical parameters were created by a MoH team.• There was not an allocated budget to conduct studies that define the technical parameters.• Peru should not follow the tobacco route and instead, it should have an impartial position because the food industry is Peruvian and should not be harmed.• The proposed technical parameters seek to gradually improve ultra-processed products, aiming at the technical parameters to be proposed by WHO/PAHO in the future.• Drastic decreases in sugar or fat in ultra-processed foods could denaturalize some of these products.• These parameters aimed to gradually modify the social palatability of Peruvians. | • **Design of the 1st draft of the regulation:**<br>• This draft was severely criticized and was not approved. |
| | • **Design of the 2nd draft of the regulation:**<br>• There is no scientific evidence that the parameters will be beneficial• The CODEX standards should be followed, since they do have evidence. | • **Design of the 2nd draft of the regulation:**<br>• The Regulation should include the parameters of PAHO (Nutrient Profile Model) because that was the indication in the Law• The implementation process could be gradual. | • **Design of the 2nd draft of the regulation:**<br>• This draft was not approved. |
| | • **Design of the approved regulation:**<br>• There is no scientific evidence that the PAHO technical parameters will be of benefit, so the CODEX standards, which do have scientific evidence, should be followed.• The industry offers to fund a study of the nutritional situation in Peru in order to develop its own technical parameters. | • **Design of the approved regulation:**<br>• The Regulation should include the parameters of PAHO (Nutrient Profile Model) because that was the indication in the Law. | • **Design of the approved regulation:**<br>• The Ministry of Health places the technical parameters of the Chilean regulation divided into 2 phases so that the Regulation can be approved by the Presidency of the Council of Ministers. |
| | **Approved Regulation:**<br>• The Law has been violated by approving a Regulation with technical parameters that were not those of WHO/PAHO and that were more flexible.• The College of Nutritionists of Peru files a lawsuit against the Ministry of Health requesting the annulment of the regulation. | **Approved Regulation:**<br>• The Regulation is legal because they followed the WHO recommendations that indicates that the MoH can include successful models, as is the case of the Chilean model.• The technical parameters of the Regulation are going to have the same amount of ultra-processed foods with nutritional warnings as with the PAHO Nutrient Profile Model.• The goal is to reach the technical parameters recommended by PAHO.• A Regulation is prepared with these parameters because it was the only way for it to be approved by the Presidency of the Council of Ministers.• The technical parameters are supported by the PAHO representative because he considered that this was the only solution. | **Approved Regulation:**<br>• The College of Nutritionists of Peru had to give up its position against the Regulation because they preferred to continue with it rather than paralyze the process of implementing the Law. |

(*Continued*)

**Table 5.** (Continued)

| Topic | Arguments against | Arguments in favor | How it was resolved |
|---|---|---|---|
| **Relevance of the other components of the Law (healthy canteens, nutrition education, etc.)** | **Design of the Law:**<br>• The government is infringing the populations' freedoms, it is going to tell parents what to buy.• The problem is not the diet but the lack of physical activity.• The problem in Peru is anemia, not obesity.• The Law does not focus on restaurants and fast food.• The solution is not to prohibit but educate. | **Design of the Law:**<br>• Overweight and obesity in children and adolescents must be addressed because it is a serious problem.• The Law is not prohibiting the sale of any food.• Children and adolescents are a vulnerable population.• This Law seeks to protect children and adolescents. | **Design of the Law:**<br>• Several articles of these components are eliminated or modified, such as the elimination of the supervision of nutritionists in the feeding of school children. |

The Table shows the main conflictive issues during the design and approval of the nutritional warnings' policy, as well as the arguments used by those in favor and against the policy and how these issues were finally solved. These conflictive issues were in regard to the regulations of publicity in the Law, the parameters during the Regulation's design, and the nutritional warnings' design during the Manual.

proposed in Peru in 2014 (Table 4). In February 2016, WHO and PAHO published the Nutrient Profile Model [39, 45], which were much stricter than all the previous ones. In face of this new evidence, the MoH replaced its Regulation of 2015 and used the parameters of the Nutrient Profile Model for the second draft, pre-published on July 25th, 2016, [42] days before President Humala finished his ruling period (28th July, 2016).

The parameters for trans-fat were published separately, on July 26th, 2016 [43], as they were prepared by a different MoH group who had already defined them in 2012 [75]. The trans-fat pre-publication established their elimination in any processed product using partial hydrogenation after 54 months. Interviewees from different sectors agreed that even though the food industry questioned if that zero trans-fat would not be technically feasible, they were mainly in favor of the initiative, possibly explaining why these parameters were easy to approve and are still in force.

"Trans-fat is the most harmful fat, much more harmful than saturated fat. Therefore, we, of course, in the formulation of our products, we use zero trans-fats." (Industry representative, IND-01)

The new government's MoH, under the presidency of Kuczynski (2016–2018), received several queries, suggestions, and critiques for the second Regulation's draft. The MoH hired a group of consultants to attend to these comments and prepare a new Regulation. The parameters (based on the WHO/PAHO Nutrient Profile Model) were one of the main conflictive issues among the Cabinet of Peru, who had to approve the Regulation (Table 5).

"The Minister [of Health] returned several times with the Regulation in hand saying: 'No, no, this does not pass in the Cabinet. The Minister of Industry, the Prime Minister, they don't want it, they say no.'" (International organization's representative, INT-01)

After the "Pura Vida" scandal, and aiming to overcome the Cabinet's resistance, the MoH adopted the Chilean parameters, and the Regulation was rapidly approved, in June 2017, two weeks after the scandal. Civil society advocates initially rejected this approved Regulation for not using the WHO/PAHO parameters (see Table 5), but then they accepted them and decided to advocate for the Manual's design.

"We reached a tacit agreement of accepting the Regulation with the parameters of Chile, right? Because we met at the College [of nutritionists] and we said: 'If we fight the

parameters of PAHO, this [the Regulation] will take even more time'; and, the fact that it takes longer, is going to imply that the other side–those against The Healthy Eating Law– would take advantage of it to create this kind of poorly made laws." (Civil society advocate, SOC-03)

**3.2.3 Proposed nutritional warnings in the period of designing the Regulation.** None of the Regulation's drafts specified a nutritional warnings' design. During 2014 and 2015 advocates proposed the traffic lights design, which were the most well-known design at the time. However, based on the Ecuadorian experience, they discarded this design because it was not easy to interpret and because it enabled the food industry to take advantage of their colors to merge them into colorful labels.

"And we also had evidence that the other alternatives, like the traffic lights, were absolutely distracting and that they didn't go to the essence of the problem. Let's say, if we have a product with two orange balls, one green ball, and two red balls, who won? What is your conclusion? I mean, the interpretation is complex. And that was clear in Ecuador which was the first [country] to use this design." (International organization's representative, INT-01)

During 2015 onwards, the octagon design gained more support among Law advocates. In Chile they showed positive results in helping consumers decide [76] and not having severe effects on the food industry's employment [77]. Likewise, the Peruvian MoH conducted small qualitative studies and found positive results with the octagons [9]. The MoH tried to specify the octagons as the selected design for the nutritional warnings in the Regulation draft published in 2016. However, opponents at the Cabinet (e.g., Ministry of Economy) argued that the Law did not mention using iconography for the warnings, so that the design could not be mentioned in the draft. Finally, opponents' argument prevailed, and the published draft did not mention any specific design (see Table 5).

Throughout these years, opponents from the food industry and Congress were against the octagons and proposed the traffic lights or Guideline Daily Amounts (GDA) designs.

"At some point we wanted to implement it (the traffic lights) in Peru because it seemed to us easier to understand than (the octagons) (. . .) (Additionally,) you can have High in sugar, yeah, it's High in sugar (but) how high in sugar is it? Is it (a product) (from) someone who made an effort to reformulate the product and instead of 5 grams it has 5.25 or 6 of sugar, versus another one who simply used the High in Sugar label and has 20 grams of sugar per product?" (Food industry representative, IND-02)

**3.2.4 Main challenges faced when trying to establish the nutritional warnings' parameters in the Regulation.** Main challenges during this period were: 1) the absence of parameters issued by WHO/PAHO until 2016; 2) the strong opposition of the food industry, media, congressmen, and members of the Cabinet (e.g. Ministry of Economy, Ministry of Production) to the different parameters and nutritional warnings' design proposed by the MoH; 3) the scarcity of local research to support the MoH's decisions, due to lack of funding and no alliances with researchers; 4) power differences between those in favor and against (i.e., the food industry had plenty of resources to pay several lawyers to confront the Regulation's drafts, whereas the MoH only had a small legal team involved in several other policies); and finally, 5) political will to assure the Regulation's approval.

### 3.3 In 2018, the Nutritional Warnings Manual was approved, defining the octagons as the selected design

**3.3.1 Summary of how the Manual was designed and approved.** The MoH pre-published a first draft in August 2017, two months after the Regulation's approval [52], aiming to approve it by November 2017. However, in September 2017, the Minister of Health was removed, and the next two Ministers did not prioritize this policy.

> "You know how many times I have been told to put away the Manual? 'Save your Manual, that's never going to come out. Save it'. Then another [Minister] came, another, another Minister, and another director. And I took [the Manual] out again! (Laughs). People told me: 'Don't bother, it won't have a chance, I don't know why you're insisting (. . .)' And I looked for an audience to tell what the warning's Manual was, yeah, until one [Minister] finally listened to me" (Decision maker at the MoH, DEC-04)

On the other hand, the new Congress, elected in July 2016 and with 60% of its congressmen from *Fuerza Popular*, the main opponent political party in the previous parliament, continued discussing bills to modify the Law and its Regulation. This was a very stagnant period for the policy, in which the MoH did not work on the Manual, and the Congress proposed changing the octagons to traffic lights or GDA and replacing the nutritional warnings for nutritional tables. As shown in Table 3, six bills from different parties were prepared during 2017 [48–51, 53, 54] and then merged into one single document (Dictum). With majority of votes from *Fuerza Popular*, the Dictum was approved by the Congress on March 9th, 2018, and sent to the country President, Kuczynski, to either approve or veto the Dictum by April 9th, 2018.

On March 21st, however, a political crisis led President Kuczynski to resign [66], without having replied to the Dictum. On March 23rd, 2018, his vice-president, Vizcarra, assumed the presidency and days later he vetoed the Dictum, arguing that the MoH considered the octagons easier to understand and that the parameters proposed by the Congress (GDA) were based on a diet for an adult person (2000 kcal) and not children or adolescents [78].

> "[President] Vizcarra arrived and the first thing he heard was: 'Sir, observe the Law'. And I'm sometimes pessimistic. I said: 'How's this man going to observe the Law, he's just coming in, he doesn't even know what it is, ahh, no, he is going to promulgate it, he won't fight with the congressmen'. But [fortunately] he observed the Law!" (Civil society advocate, SOC-01)

The new Minister of health returned to the draft pre-published in August 2017 and the Manual was finally approved in June 2018 [9]. The opposition from the Congress majority did not continue because *Fuerza Popular*, the main opponent political party was under public scrutiny due to denounces of corruption [79].

> "They [Fuerza Popular] cut off arms and legs to the Law, [but now] they wanted to implement their colored GDA. And they had the votes to do it. And [you can] remember that the president observed it [the Dictum]. That draft returned [to the Congress], and they [Fuerza Popular] had the votes to insist. If they would have insisted on [modifying] the Law, this would be history, we would not be talking about this right now." (Congressman 2011–2016, POL-02)

Other interviewees argued that the confrontation between the MoH and the Congress reached the public opinion, and a dichotomy of health vs economy was created, where "the

powerful food industry" was seen as only seeking to profit disregarding people's health. Others added that a popular doctor who had a program in the biggest radio station (Dr. Elmer Huerta) was a key advocate to easily explain to the public the evidence supporting the octagons and to highlight the economic interests behind the proposals from the Congress. PAHO and civil society advocates were also strong supporters of the octagons.

> "Something that worked very well was having Elmer Huerta as a champion, right? (. . .) He [normally] tries to be fairly impartial on political issues; but, on this one, he wasn't. And he was very clear in pointing out who were against the Law." (Independent researcher, INV-03)

Once the Manual was approved, the nutritional warnings' first implementation's phase began one year later, on June 17th, 2019; and its second phase, on September 17th, 2021.

**3.3.2 Proposed nutritional warnings in the period of designing the Manual.** During these years, the nutritional warnings design was the main conflictive issue. Those in favor of the Law advocated for the octagons, arguing that they were easy to understand, and clear enough to make rapid decisions [80]. Opponents proposed the traffic lights, GDA or a mix of both, arguing that consumers needed more information (i.e., quantity of nutrients) than solely short messages.

> "We ruled based on what the consumer law establishes, that a consumer must have all the appropriate information for decision making. So, that's why we went for a mixed figure between the tables of nutritional values (. . .) and combine it with colors, or messages to inform if the minimum allowed was exceeded, according to the table established by the Ministry of Health." (Congressman, 2016–2018, POL-03)

Based on qualitative studies conducted by the MoH, but mainly due to discussions at the Cabinet, the octagons' design changed from the 2017 draft to the Manual approved in 2018. For instance, their colors changed from red to black, the label "Ministry of Health" was removed, and the minimum package's size that could be labelled changed from 20 cm$^2$ to 50 cm$^2$ [9].

> "The warnings' Manual is basically about technical aspects, but there were also negotiations. No, no, they didn't call it 'negotiations' but 'decisions' at the Senior Management [cabinet], more political, let's say, right? In other words, not everything was finally solved by the technical team. The technical team had a position, but the ultimate decision was at the Senior Management level" (Decision maker at the MoH, DEC-04)

**3.3.3 Main challenges faced when trying to establish the nutritional warnings' design in the Manual.** Main challenges in this period were 1) the strong power of the opponents at the Congress-level, risking the nutritional warnings policy; 2) the strong opposition to the octagons design by the Cabinet, leading to delays in the Manual's approval and changes to the MoH's proposed design; 3) the scarce political will of two health Ministers to finish on the Manual (Sept 2017 –Apr 2018); 4) political instability, which led to changes of the country President; 5) the limited locally-generated research to inform the MoH's decisions regarding the warnings' parameters and design, which was used by opponents as an argument to delegitimize the data from other countries that the MoH presented as evidence.

**3.3.4 Why this policy was approved: Analysis using the Kaleidoscope Model.** Following the Kaleidoscope Model [18], which aims to explain policy changes, we conclude that the nutritional warnings policy was successfully issued because several hypotheses were addressed

in ways that encouraged policy-supportive agenda setting, design, and adoption, as shown in Table 6.

## 4. Discussion

This study describes, over the 2012–2018 period, the milestones and key stakeholders' roles and stances during the Peruvian nutritional warnings policy design and approval, and analyzes the policy drivers that made it possible. The approval of the black octagons as the Peruvian nutritional warnings was a lengthy and sinuous process that faced strong opposition from the media and food industry representatives, as well as members of the different Congresses and Governments. On the contrary, civil society organizations and PAHO were strong advocates for the policy, and most health Ministers and Presidents supported its approval. After all the challenges faced, Peru successfully defined the black octagons as the nutritional warnings' design, which has proven to be easier to understand by consumers [81] and appears to produce the largest public health benefits [82].

Regarding key stakeholders' roles and stances, we found that they came from different sectors and played relevant roles in favor or against the nutritional warnings policy along the years. The main opponents throughout the process were the media and food industry, which have proven to be major opponents of nutritional warnings policies in Latin American countries for its impact on publicity, such as Chile [83], Uruguay [12], Colombia [84, 85], and Brazil [86]. The industry arguments in Peru were similar to those used in other Latin American countries when pursuing their nutritional warnings policies. For instance, critiques towards the evidence used by promoters to support the parameters and warnings' design, arguing that it was too weak [12], that the food industry's proposals were not included [86], or that this policy did not show reductions in obesity in other countries [85]. Another common argument described the economic loss that the warnings' implementation would bring to their countries, such as loss of jobs [12, 83, 86], rise of prices [12], or violations to international trade agreements [83, 86]. None of these potential risks, however, occurred in the countries that have implemented the nutritional warnings [83], including Peru [87]. Importantly, beyond their arguments, we found that some powerful stakeholders were aligned to the food industry stance and acted against the warnings policy, such as congressmen and policymakers' at ministries of economic sectors, as seen in Uruguay [12], Chile [83], and Colombia [85], which risked its approval and further implementation by continuously changing the policy contents and, potentially, their intended results [10].

On the other hand, civil society organizations were vital to advocate for this policy by doing public appearances on TV and radio to inform about the policy and demanding its rapid approval and implementation. Similar to Colombia, these organizations promoted the bill to approve the nutritional warnings and the prohibition of foods with FoP labels to children and adolescents [85]. In clear contrast to Peru, academics in other countries, such as Chile [88], Mexico [15], and Uruguay [12], worked closely with policymakers and civil society advocates, supporting their arguments against opponents to ease the approval of their policies. Importantly, this reflection arises within a context and a policy environment where local support to science in the country is generally limited [89], even when the use of research could ease policymakers' decision by broadening their options, anticipating potential risks and benefits, and understanding why some policies work and others fail [90]. Moreover, these findings are important to anticipate in other settings where similar policies are yet to be pursued and where simultaneous support to the generation of local evidence will be essential to protect and progress with related societal prevention approaches.

**Table 6. Evidence addressing hypotheses of agenda setting, design, and adoption processes for the nutritional warnings' policy.**

| Determinant of policy change | Evidence addressing hypotheses of agenda setting, design, and adoption processes |
|---|---|
| **AGENDA SETTING** | |
| Recognized, relevant problem | • Unhealthy eating decisions among children and adolescents was unanimously seen as a relevant problem. |
| Focusing events | • The "expert consultation" organized by PAHO in 2011<br>• The "Pura Vida" scandal in 2017<br>• The promising results of using octagon warnings in Chile<br>• The support of key health Ministers who pursued the policy approval<br>• The political context during the Manual's approval |
| Powerful advocates | • The policy was supported by strong individuals and institutions from the Congress, civil society organizations, Ministry of Health, and PAHO.<br>• Even though supporters had some important discrepancies (See Table 5), they later agreed on joining efforts and move the policy forward. |
| **DESIGN** | |
| Knowledge and research | • The PAHO's Nutrient Profile Model was not published until 2016, three years after the Law's approval;<br>• Most studies were small-scale only and conducted by reduced groups at the MoH or civil society organizations (Table 3);<br>• No alliances with researchers to generate or support local evidence. |
| Norms, biases, ideology, and beliefs | • When drafting the Law, neither the policy advocates nor opponents proposed any specific design in terms of shape or colors, which delayed the approval of the octagons.<br>• Drafts were constantly edited based on critics from opponents, who used controversial evidence that was not recommended by health authorities (i.e. daily reference values, traffic lights design). |
| Cost-benefit calculations | • Several negotiations yielded elements that weakened the regulations (i.e. not using the strictest parameters or changing the minimum package's size), but that ultimately allowed them to be approved.<br>• Conflictive components were commissioned to a subsequent policy document (e.g. the parameters to the Regulation, and the warnings' design to the Manual) to assure a prompt approval.<br>• The nutritional warnings' purpose was to bring health gains at the population level (i.e., use of services' savings, DALYs), especially for children and adolescents. |
| **ADOPTION** | |
| Powerful opponents vs proponents | • Even though opponents in the Congress and food industry were so powerful that they risked the policy documents' approval several times, and had a larger economic capacity in contrast to the scarce resources of the proponents, most Presidents and Health Ministries supported the policy. |
| Government veto players | • Government agents with ultimate decision-making power, such as Presidents and Health Ministers, were either in favor or neutral to the policy, but never against it, which enabled its continuation.<br>• However, powerful veto players at the Congress and within the Cabinet, related to the economic interests of the food industry, jeopardized the policy several times by trying to modify the Law or by not approving the Regulation and Manual, especially during the political instability of 2016–2018. |
| Propitious timing | • The opposition of the food industry and media against the regulations of publicity, was a propitious time to include the nutritional warnings in the Law.<br>• The "Pura Vida" scandal received local attention and was used to advocate for the Regulation's approval.<br>• The political instability in the country and the public scrutiny that the main opposing party in the Congress received, seemed to be the reasons why the Congress stopped their efforts to change the Law and its Regulation, yielding the path for the approval of the Manual. |

Another relevant finding is that beyond the technical information needed to define the warnings' design and parameters, the policy process of the nutritional warnings highly depended on political issues, such as the political will of key decision makers in Congress and

the Cabinet and the political stability in the country. Indeed, Peru was not the only country whose political contexts eased the approval of the nutritional warnings' policy. In Mexico, the change of government in 2018 began a "fight against corruption" and public shaming for decision makers who protected the food industry and not consumers which, similar to the dichotomy of health vs economy created in Peru, enabled the progress of their warnings' policy [15]. Thereby, this policy exemplifies the relevance of the political context during the design and approval of a health policy, which is usually ignored in academic studies [91].

Finally, the Kaleidoscope Model allowed us to identify the main strengths and weaknesses during the policy design, as well as the main drivers for its successful approval. As previous research using the Kaleidoscope Model in Latin America shows, we found that the media and food industry are key players in the policymaking process and should be included in the model [92]. Yet, even though the policy approval took several years, it was ultimately adopted because key government veto players (e.g., Health Ministers, Presidents) were mostly in favor of the policy, and despite having some powerful and influential opponents (e.g., Congressmen, Cabinet, food industry), the capacity of supporters proved to be stronger.

## 4.1 Implications for public policies

The nutritional warning's policy in Peru was the result of several modifications to its technical contents and negotiations between relevant stakeholders. Important strengths of the policy include its current implementation despite the challenging environments it faced, its use of black octagons, and the fact that Peru is one of the few countries in Latin America, with Mexico and Venezuela, whose policy includes warnings for industrially produced trans-fat that aims for its elimination, as recommended by PAHO [93]. However, the policy still has room for improvement, for example, by incorporating other nutrients to be avoided, such as sweeteners and caffeine, expanding the minimum label size for the warnings, mandating the inclusion of products' nutritional information, and using the PAHO's Nutrient Profile Model for the warnings' parameters [4]. Actually, in January 2022, the Peruvian Supreme Court demanded that the government use the PAHO Nutrients Profile Model's parameters–stricter than the current ones as the Law stated [94]. In July 2022, the MoH presented a draft to include them in the Regulation [95].

## 4.2 Strengths and limitations

This study conveys the experiences and perspectives of diverse key supporters and opponents involved in the design and approval of the nutritional warnings' policy in Peru, which provides a deeper insight of the nature of this process and triangulates their different perspectives into a solid analysis. Likewise, including the most relevant policy documentation and news about the nutritional warnings' design and approval in our study enabled a better contextualization and analysis of our interviewees discourses. One limitation is that the elapsed time between the policy design and the interviews, could potentially lead to a recall bias. However, discourses were triangulated among informants as well as with the policy documents and news.

## 5. Conclusion

This study describes and analyzes the lengthy and winding design process of the nutritional warnings policy in Peru, from 2013 to 2018. The stances of some stakeholders changed along the years, but in general, this policy was mainly supported by some congressmen, policymakers from the health and social sector, civil society advocates, and PAHO; and usually opposed by the food industry and media representatives, as well as some civil society organizations and policymakers from the economic sector. The parameters and the warnings' designs changed

frequently across the years, with different proposals in each draft, reflecting the disagreement among supporters and opponents, and the pressures from the industry. These changes led to important losses, such as using more flexible parameters than those proposed by PAHO or not using the octagons in small labels, however, they were perceived as necessary compromises to get the approval of the policy. Based on the Kaleidoscope Model, we identified the main strengths and weaknesses during the policy design, as well as the main drivers for its approval. Acknowledging that the nutritional warnings policy is being implemented in different Latin American countries [4], these results will be useful for advocates to learn from the experience of Peru and anticipate potential difficulties and ways to overcome them.

## Supporting information

**S1 Text. Code-book.**
(DOCX)

## Author Contributions

**Conceptualization:** Francisco Diez-Canseco, Victoria Cavero.

**Formal analysis:** Victoria Cavero, Juan Alvarez Cano.

**Funding acquisition:** Francisco Diez-Canseco.

**Investigation:** Victoria Cavero, Lorena Saavedra-Garcia.

**Methodology:** Francisco Diez-Canseco, Victoria Cavero, Juan Alvarez Cano, Lindsey Smith Taillie, Francesca R. Dillman Carpentier, J. Jaime Miranda.

**Project administration:** Francisco Diez-Canseco, Lorena Saavedra-Garcia.

**Supervision:** Francisco Diez-Canseco, Lindsey Smith Taillie, Francesca R. Dillman Carpentier, J. Jaime Miranda.

**Validation:** Lorena Saavedra-Garcia.

**Writing – original draft:** Victoria Cavero, Juan Alvarez Cano.

**Writing – review & editing:** Francisco Diez-Canseco, Victoria Cavero, Juan Alvarez Cano, Lorena Saavedra-Garcia, Lindsey Smith Taillie, Francesca R. Dillman Carpentier, J. Jaime Miranda.

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
