## [Decision Letter · Decision Letter 0]

25 Nov 2022

PGPH-D-22-01451

Design and approval of the nutritional warnings’ policy in Peru: Milestones, key stakeholders, and policy drivers for its approval

Dear Dr. Francisco Diez-Canseco,

Thank you for submitting your manuscript to PLOS Global Public Health. After careful consideration, we feel that it has merit but does not fully meet PLOS Global Public Health’s publication criteria as it currently stands. Therefore, we invite you to submit a revised version of the manuscript that addresses the points raised during the review process.

We look forward to receiving your revised manuscript.

Kind regards,

Roopa Shivashankar, MD, MSc

Academic Editor

Journal Requirements:

2. In the online submission form, you indicated that "Data cannot be publicly shared because it contains potentially identifying information. Since we interviewed key stakeholders with specific and potentially unique knowledge about the nutritional warnings policy in Peru, the information they provided could ease their identification. Data requests could be sent to the corresponding author and they could be accessed on reasonable request" journals now require all data underlying the findings described in their manuscript to be freely available to other researchers, either 1. In a public repository, 2. Within the manuscript itself, or 3. Uploaded as supplementary information.

Additional Editor Comments (if provided):

Reviewers' comments:

Reviewer's Responses to Questions

**Comments to the Author**

1. Does this manuscript meet PLOS Global Public Health’s publication criteria? Is the manuscript technically sound, and do the data support the conclusions? The manuscript must describe methodologically and ethically rigorous research with conclusions that are appropriately drawn based on the data presented.

Reviewer #1: Yes

Reviewer #2: Yes

2. Has the statistical analysis been performed appropriately and rigorously?

Reviewer #1: N/A

Reviewer #2: Yes

3. Have the authors made all data underlying the findings in their manuscript fully available (please refer to the Data Availability Statement at the start of the manuscript PDF file)?

Reviewer #1: Yes

Reviewer #2: No

4. Is the manuscript presented in an intelligible fashion and written in standard English?

Reviewer #1: Yes

Reviewer #2: Yes

5. Review Comments to the Author

Reviewer #1: Thank you for the pportunity to review this paper. It deals with a very important topic and provides insights into a country’s journey in bringing about the warning labels to discourage unhealthy food consumption and promote informed healthy food choices. This work has some important takeaways for several LAMI countries which are in the process of bringing out FOPL regulations.

The methodology is used appears appropriate to document the key milestones, policy drivers and hindrances.

I have a few questions and suggestions

The paper in its current form appears too long and difficult to read.

This needs to considerably shortened while keeping it to adequate length so as not to lose the flavour of the qualitative work that informs the outcomes.

In its current form the paper has too much of narrative and am afraid it will be difficult to engage the readership

I suggest the authors to organise the material to document the key milestones, key drivers, hindrances, how they were overcome. With appropriate sub-heads and second level subheads

English language editing is required and the authors if they are not native English speakers may seek help if needed

L54: Indicates that increasing childhood obesity in Peru is the reason for bringing FOPL (Warning labels) regulation. Is it so? Was rising obesity and NCDs among people of all age groups not a reason?

L67 delete ‘occur’

L68-70: Not clear

‘Context about the nutritional warnings policy in Peru’ this part is a misfit in the methods section. This should either be in intro or in results

Methods

Not clear how the participants were identified?

Why 25 Key informants were conducted? Why not 26 or 27 0r 40?

Was media not a key stakeholder group?

How about consumer organisations?

Not clear how masters’ students among researchers were considered a good fit to be called key informants? These were students of which disciplines?

L128: Not clear if all four researchers were present for every online interview or were they one-to-one speech/discussion events?

147-148 triangulation with news and policy documents analysis was an important part of methodology? More clarity is needed on how it was done?

Titles of the tables need to be self-explanatory. They are too shot and colloquial

Some of the unpublished reports have been included in the scientific events in table-3. How did you source them and were they peer reviewed or were they just studies conducted and not even documented by the ministries/regulators?

L198 – congressmen or congress?

L202-204: how do you say that the congress members were a majority compared to media and industry? Did you mean in the committees that debated the law? The views of the congress outweighed others only because they were a majority?

L303-305: This quote is from the regulator/policy maker. Are there any quotes from the opponents of this view from other stakeholder groups

The narrative in the results section is too long and the reader is likely to lose track of what is being talked about. It only appears to take a predetermined path of showing how the policy makers and politicians with commitment and opinion leaders with a strong will to take materialise the warning labels have manoeuvred through tough times. It does not adequately bring out the quotes and views of the opponents. I am not able to connect to the Kaleidoscope Model and the key stages that it talks about

To me table 5 is the key to the results. I appreciate the authors for this. But the same clarity cannot be seen in the narrative

The discussion section is a bit weak and ends up repeating the results. Some important points including the how nutrition literacy was enhanced (if at all), how such issues were dealt with in other south American countries, the importance of political will, what were the experiences with industry and media organizations in other countries where warning labels were introduced etc would help enrich the discussion section.

It is not clear what the document search and media coverage data has yielded and how it contributed to the narrative of the paper especially in documenting the timeline and milestones.

Over all, I suggest the authors to revise the paper to make it more interesting as the paper has some key take ways to offer.

Reviewer #2: It is a well written manuscript describing detailed process followed for approval of the ‘nutritional warnings’ policy in Peru. Authors have provided detailed timelines with policy decisions and action points. The following details may be specified to add more clarity to the manuscript.

Introduction

Line 47-50: Authors are requested to provide reference for processed and ultra processed foods.

Participants:

Line 106 - 107: It is not clear, how was the sample size of 25 participants calculated? The basis for this should be provided.

104 - 112: The authors selected and interviewed 25 informants who were closely involved in the design, implementation and/or monitoring of the nutritional warnings policy in Peru. Were there additional criteria for selecting the participants/ informants [e.g. years of experience; current job role etc.], if so, the same should be specified.

Line 107 - 108: The way 'policy makers' and 'politicians' have been defined, were there any overlaps between the two? If so the same should be specified.

Table 2: Authors have defined Informants in 5 categories [line 107-109]; Category 2 in table 2, 'Members of Congress' should allign with the category specified in text [category 2) politicians; line 108]

Line 145 - 146: Please provide reference where the 'implementation and audit process' has been specified

Line 207; Extracts of the interviews should be labelled according to the 5 categories of informants defined in lines 107 - 109.

Section 9; The sequence of hypothesis appearing under 'analysis of Kaleidoscope Model' appears randomly and does not follow numerical sequence. If feasable, the hypothesis should appear in numerical sequence for better undrstanding and clarity.

6. PLOS authors have the option to publish the peer review history of their article (what does this mean?). If published, this will include your full peer review and any attached files.

**Do you want your identity to be public for this peer review?** For information about this choice, including consent withdrawal, please see our Privacy Policy.

Reviewer #1: **Yes: **SubbaRao M Gavaravarapu

Reviewer #2: **Yes: **Swati Bhardwaj

---

## [Decision Letter · Decision Letter 1]

23 May 2023

Design and approval of the nutritional warnings’ policy in Peru: Milestones, key stakeholders, and policy drivers for its approval

PGPH-D-22-01451R1

Dear Dr Francisco Diez-Canseco, 

We are pleased to inform you that your manuscript 'Design and approval of the nutritional warnings’ policy in Peru: Milestones, key stakeholders, and policy drivers for its approval' has been provisionally accepted for publication in PLOS Global Public Health.

Best regards,

Roopa Shivashankar, MD, MSc

Academic Editor

Reviewer Comments (if any, and for reference):

Reviewer's Responses to Questions

**Comments to the Author**

1. If the authors have adequately addressed your comments raised in a previous round of review and you feel that this manuscript is now acceptable for publication, you may indicate that here to bypass the “Comments to the Author” section, enter your conflict of interest statement in the “Confidential to Editor” section, and submit your "Accept" recommendation.

Reviewer #1: All comments have been addressed

2. Does this manuscript meet PLOS Global Public Health’s publication criteria? Is the manuscript technically sound, and do the data support the conclusions? The manuscript must describe methodologically and ethically rigorous research with conclusions that are appropriately drawn based on the data presented.

Reviewer #1: Yes

3. Has the statistical analysis been performed appropriately and rigorously?

Reviewer #1: N/A

4. Have the authors made all data underlying the findings in their manuscript fully available (please refer to the Data Availability Statement at the start of the manuscript PDF file)?

Reviewer #1: No

5. Is the manuscript presented in an intelligible fashion and written in standard English?

Reviewer #1: Yes

6. Review Comments to the Author

Reviewer #1: The authors have adequately addressed all my comments. I have no further queries/comments

7. PLOS authors have the option to publish the peer review history of their article (what does this mean?). If published, this will include your full peer review and any attached files.

**Do you want your identity to be public for this peer review?** For information about this choice, including consent withdrawal, please see our Privacy Policy.

Reviewer #1: **Yes: **SubbaRao M Gavaravarapu
